# Prevalence and risk factors of childhood diarrhea among wastewater irrigating urban farming households in Addis Ababa

Adane Sirage Ali[1,2]*, Sirak Robele Gari[1], Michaela L. Goodson[3], Claire L. Walsh[4], Bitew K. Dessie[5], Argaw Ambelu[1]

1 Division of Water and Health, Ethiopian Institute of Water Resources, Addis Ababa University, Addis Ababa, Ethiopia, 2 Department of Urban Environmental Management, Kotebe University of Education, Addis Ababa, Ethiopia, 3 Newcastle University Medicine Malaysia, Newcastle upon Tyne, Malaysia, 4 School of Engineering, Newcastle University, Newcastle, United Kingdom, 5 Water and Land Resource Centre, Addis Ababa University, Addis Ababa, Ethiopia

* adane.sirage@aau.edu.et

**Data Availability Statement:** All relevant data are within the paper and its Supporting Information files.

## Abstract

### Introduction

Childhood diarrhea is one of the major contributors to the morbidity of under-five children in Ethiopia. Although researchers determine the risk factors varyingly, the exposure route to the pathogens is usually complicated. This study aims to investigate the prevalence and risk factors of diarrhea among children under the age of five among wastewater irrigation farming households in Addis Ababa, Ethiopia.

### Methods

Cross-sectional study was conducted among 402 farming households from November 2021 to February 2022. Data was collected using a face-to-face interviewer-administered questionnaire. Stata version 14 software was used to analyze data. Factors associated with the prevalence of diarrhea was identified using binary logistic regression. Multivariable analysis was carried out to determine an adjusted odds ratio at a confidence level of 95% and level of significance at 0.05.

### Results

The overall prevalence of under-five children diarrheal cases was 22.3%. The odds of diarrhea are associated with a multitude of variables. Major wastewater-related determinants associated with diarrhea are body washing with irrigation water [AOR: 37.7, 95%CI (3.1, 358)], contaminated cloth with irrigation water [AOR:10.8,95%CI(0.6, 205)], use of protective clothing during farm work [28.9,95%CI (3.9, 215)], use of farm work cloths at home [AOR: 31.7, 95%CI (4.4, 226)], and bringing unwashed farm tools to home [94 (5.7, 1575)].

**Funding:** Name of the author who received the grant - Adane Sirage This work was supported by the Water Security and Sustainable Development Hub which is funded by the UK Research and Innovation's Global Challenges Research Fund (GCRF), Grant no.: ES/S008179/1. The funding organization does not play any role in the study design, data collection and analysis, decision to publish, or preparation of the manuscript.

## Conclusion

The high prevalence of under-five children diarrheal disease among wastewater irrigation households was strongly associated with factors related to occupational exposure. Thus, to decrease childhood diarrheal among urban agriculture farmers, appropriate precautions need to be taken.

## 1. Introduction

Irrigation using wastewater is becoming common practice among urban farmers worldwide [1]. The continuous supply of wastewater enables urban farmers to cultivate year-round [2]. Besides its perennial flow, the high nutrient content of wastewater minimizes fertilization needs and consequently reduces input costs for crop growth. However, wastewater also contains a multitude of pathogens, parasites, and potentially harmful chemicals, particularly when it is used untreated. Among these health hazards, the diarrheal disease remains one of the most prevalent environmental health problems in water resource-poor countries [2, 3]. Diarrhea is responsible for the deaths of more than 90% of children under-five years old in low and lower-middle-income countries; regionally, South Asia and Sub-Saharan Africa accounted for the majority (88%) of deaths in the same age group [4]. In Ethiopia, acute diarrhea is one of the major contributors to the morbidity of children under-five [5]. The 2016 Ethiopian DHS report showed a 12% prevalence of acute diarrhea at the national level [6]. Approximately 90% of diarrhea disease occurs due to poor sanitation, lack of access to clean water supply, and inadequate personal hygiene, all of which can be easily improved by health promotion and education [7]. Diarrhea is a leading cause of morbidity and mortality in children. It can be defined as the passage of three or more loose or liquid stools daily [8].

In wastewater irrigation areas, the factors contributing to the occurrence of diarrhea are dynamic. The association between wastewater use and public health risks has been assessed in various countries such as Israel, Morocco, Mexico, and Pakistan, where wastewater is commonly used for farm irrigation [9, 10]. High-risk groups of people for these diseases are farmers with prolonged wastewater contact, their families, and nearby communities exposed to wastewater irrigation, and consumers of the product grown [3].

In Addis Ababa, using highly polluted river water for horticulture irrigation has a long tradition. The condition of the river fulfills the definition of wastewater [11]. Several studies show the poor microbiological quality of the irrigation water [12, 13], and the irrigation products, particularly vegetables [14]. However, apart from reporting *E.coli* and coliform numbers in the polluted river water and fecal contamination of vegetables, none of them further investigated the route of exposure and associated consequences. To date, no study has been conducted considering the public health concerns of microbiological contamination of the irrigation water and the products. Although there are plenty of studies dealing with diarrhea in the city [7, 15], none studied wastewater irrigation farming households.

Though the presence of fecal coliforms in wastewater-contaminated river water, irrigated soil, and irrigation products is a fact, there is no clear evidence of how the complicated exposure pathways affect farming households, particularly those of farm workers and their most vulnerable family members. The magnitude and additive effects of wastewater irrigation on the odds of diarrhea among farming households, which already have poor sanitation facilities, is not clear. The importance of wastewater irrigation to diarrheal occurrence under poor sanitation is largely unknown and no studies have assessed the determinants of diarrheal diseases

associated with exposure to wastewater use in agriculture. Therefore, this study assesses the prevalence of child diarrhea and major determinants among farming households using wastewater for irrigation in Addis Ababa.

## 2. Materials and methods

### 2.1 Study area and sampling sites

A survey was carried out in Addis Ababa along the two rivers (Big Akaki and Little Akaki Rivers) crossing the city. Addis Ababa's population was estimated to be 3,860,000 of which 1,822,000 (47.2%) were males and 2,038,000 (52.8%) were females [16]. Only 64 percent of the solid waste generated in the city is properly disposed of. Approximately 74% of the residents use pit latrines, only 7% use flush toilets, and 17% use open field toilets [16]. The 2008 basic indicator assessment in the city showed that 26% of the houses and the majority of slum dwellers had no toilet facilities, 33% of households shared a toilet with more than six households, 35% of the generated garbage/refuse was never collected, and 71% of the households did not have adequate sanitation facilities [17].

Like in other developing countries, polluted stream water has been used for crop production within and around Addis Ababa since the 1940s to produce a variety of crops for both the market and home consumption. More than 1240 ha of land is irrigated for vegetable production using water from the Akaki River alone, and this agricultural system supports more than 1260 farming households in the city and at across its periphery [18]. Almost all these farmers use untreated wastewater and polluted rivers, accounting for about 61% of the city vegetable supply, and 90% of the leafy vegetable supply [18]. For this study, nineteen urban farming sites (eleven wastewater-irrigation and eight non-wastewater-irrigating farming sites) along the two rivers crossing the city were chosen for data collection (Fig 1).

### 2.2 Sample size determination and sampling techniques

The sample size estimation for this study was determined based on the sample size estimation of a longitudinal study designed to investigate the incidence of diarrhea. For independent cohort design, the sample size is commonly determined by using the following formula [19]:

$$n = \frac{[Z_{1-\alpha/2}\sqrt{\{(1+1/m)p*(1-p)\}} + Z_{1-\beta}\sqrt{\{p_o*(1-p_o/m)p_1(1-p_1)\}}]^2}{(p_o - p_1)^2}$$

**Where,** n = total number of desired study subjects (case) to identify true relative risk with two-sided Type-I error; m = number of subjects (control) per experimental subject = 1; $Z_{1-\beta}$ = the desired power (0.84 for 80% power, and 1.28 for 90% power); $Z_{1-\alpha/2}$ = critical value and standard value for the corresponding level of confidence (at 95% CI = 1.96); $P_o$ = Possibility of an event in controls 18.12% [20]; and $P_1$ = Possibility of an event in the experimental subject = 30.8% [21]. Taking all this information and assumptions into account, the sample size of the study population was calculated. And 370 participants were included in the study. Assuming a 10% dropout rate, the total sample size was 370+ 10% dropout = 407 subjects.

### 2.3 Data collection tools and procedures

Before data collection started, all participants were asked for their willingness, and informed written consents were obtained. After registration, for each participant an ID number was used equivalent to their name, and in the rest of the study all the data were recorded by their

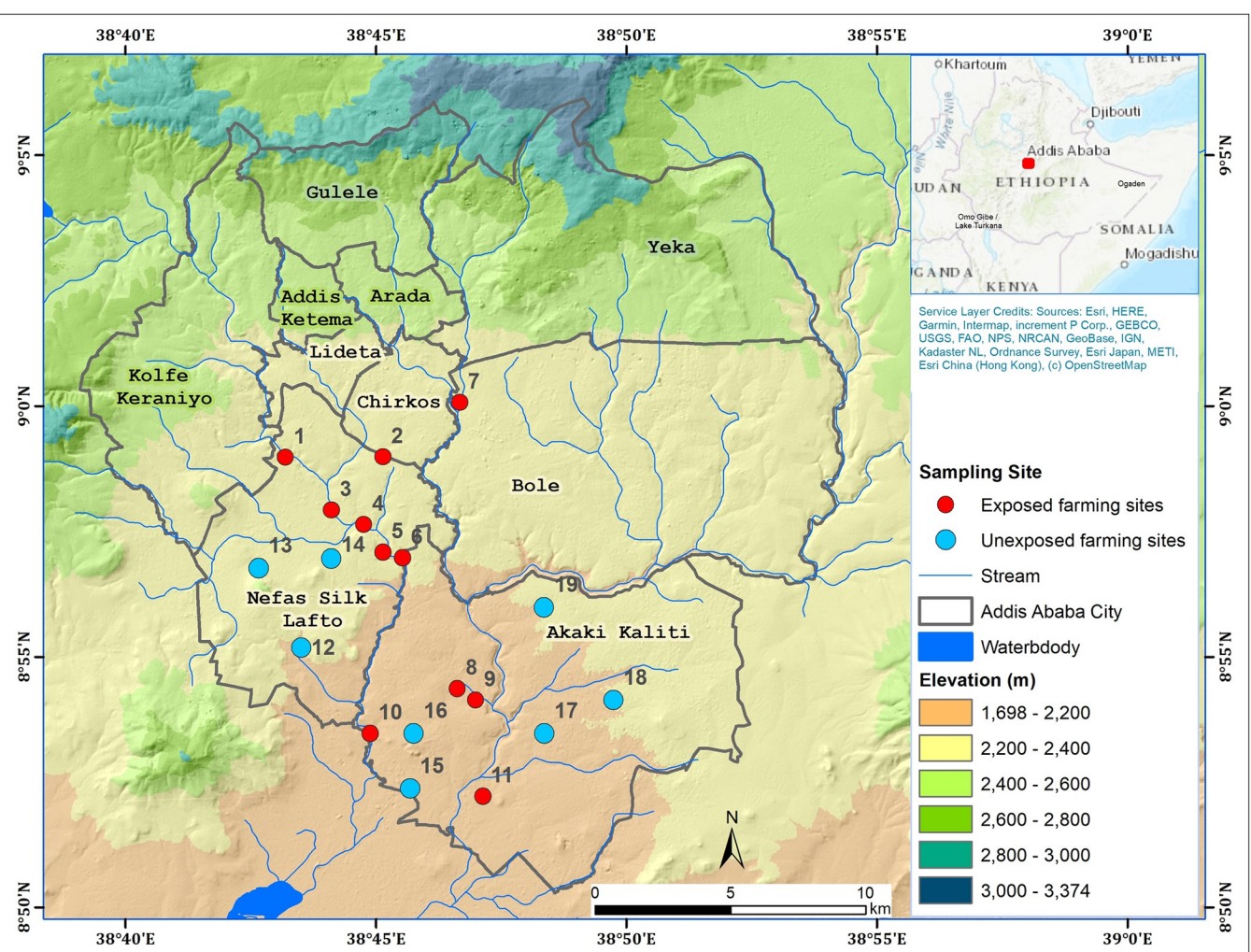

**Fig 1. Map of the study area and sampling sites (Red dots, numbers 1 to 11 are wastewater-irrigated farming sites, whereas the blue dots, numbers 12 to 19, are non-wastewater-irrigated farming sites in the labeled sub-cities).**

registered ID number which only the authors know the associated name. The participants were employed for one vegetable growing periods, from February to May 2022. A structured questionnaire was used to collect data from the households. Five percent of the households were randomly selected for a pilot study to ensure that the heads of the households and the child guardians understood the questions and the purpose of the study. Based on the pre-test evaluation, questions were revised. After training, data collectors conducted the survey by interviewing the primary farmer and child guardians using the revised questionnaire. If the guardians were not farmworkers, they answered the children-focused questions only. After checking for errors, inconsistencies and completeness, the data were entered into an excel spreadsheet and then exported into STATA software for data cleaning, verification, and analysis.

## 2.4 Data analysis

STATA Version 14.0 (Stata Corp) was used to analyze the data. Frequency tables were used to summarize the socio-demographic characteristics of the study participants, sanitation and

behavioral characteristics, occupation exposure factors, and magnitude of diarrhea. Diarrheal infection factors were determined by using bivariate and multivariate analysis. Determination of association involved estimating the crude odds ratio (COR) using bivariate analysis and adjusted odds ratio (AOR). Bivariate analysis was employed to identify factors associated with acute diarrhea at p<0.05 without controlling confounders, whereas in the multivariable analysis, the association between occupational exposure factors with acute diarrhea was examined by controlling for potential confounders [22].

The variance inflation factor (VIF) is used to assess the multi-collinearity of variables. From the AOR analysis, variables with p<0.05 were taken as statistically significant and independently associated factors with acute diarrhea.

## 3. Results

### 3.1 Socio-demographic and economic characteristics of the participants

In this study, 407 wastewater-farming households provided a response rate of 95% (386) households). Farmworkers are typically the heads of the household. Table 1 depicts the socio-demographic and economic characteristics of the study participants and their association with the odds of diarrhea among children under-five. The sex and educational level of the household head, and family size are significantly associated with the prevalence of child diarrhea. About 82% of the respondents were male, the majority over the age of 31. Children from female household heads are 4.9 times more likely to develop diarrhea than children from male household heads [(COR: 4.9 95%CI (2.5, 9.5)]. The odds of diarrhea infection among children from illiterate household heads are 2.6 times higher than among children from literate (primary to high school) household heads [COR: 2.6, 95%CI (1.3, 2.5)]. Children from better-

**Table 1. Univariate and bivariate analysis of occurrence of diarrhea and socio-demographic and economic factors (n = 386).**

| Name of variables | Category | Response | COR (95% CI) | P-value |
|---|---|---|---|---|
| Age of the household head/farmworker (years) | ≤30 | 24(6.2) | 1 | |
| | 31–40 | 123(31.8) | 1.7 (0.35, 8.2) | 0.50 |
| | 41–50 | 135(35) | 1.6 (0.3, 8) | 0.55 |
| | >51 | 104 (27) | 4 (0.84, 19.4) | 0.08 |
| Sex of household-hold | Male | 294 (76.2) | 1 | |
| | Female | 92 (23.8) | 4.9 (2.5, 9.5) | 0.000 |
| Education of household-head | No formal education | 170(44) | 2.6(1.3,5.2) | 0.007 |
| | Elementary–high school | 196(50.8) | 1 | |
| | Certificate and above | 20(5.2) | -5.3 (1.4,19.5) | 0.01 |
| Family members | 1–3 | 64(16.6) | 1 | |
| | 4–6 | 259(67) | 1.5(0.5, 4) | 0.44 |
| | > 6 | 63(16.4) | 7(2.3, 21.7) | 0.001 |
| Income (ETB) | <1000 | 122(31.6) | 2.1(0.2, 18) | 0.5 |
| | 1001–3000 | 148(38.3) | 2(0.2, 17.9) | 0.5 |
| | 3001–5000 | 108(28) | 1(0.1, 9.7) | 0.9 |
| | >5000 | 8(2.1) | 1 | |
| Number of rooms | 1–2 | 167(43.3) | 3.3 (1.2, 9.4) | 0.02 |
| | 3–4 | 193(50) | 1.9 (0.3, 11.4) | 0.5 |
| | >4 | 26(6.7) | 1 | |
| Number of people sleeping in a room | 1–2 | 160(41.5) | 1.5(0.5, 4) | 0.4 |
| | 3–4 | 163(42.2) | 1.5(0.5, 4) | 0.4 |
| | >4 | 63(16.3) | 1 | |

educated (above high school level) families had several times fewer odds of diarrhea compared to less educated and non-educated family heads [COR:5.3, 95%CI(1.4,19.5)], which is in line with several research findings [23]. About 70% of the households have 4 to 6 family members. A larger family significantly increases the odds of diarrhea compared to a small family (p<0.05).

The age of the household head, income of the household and the number of rooms in the house does not have a significant association with diarrhea occurrence. The monthly income of the majority of the households (98%) was within the range of 1000–5000 ETB. Though income does not show significant association with the occurrence of child diarrhea, it can indirectly influence other important factors such as sanitation, hygiene, and drinking water supply facilities.

### 3.2 Prevalence of diarrhea among children under the age of five

The diarrheal disease among under-five children was assessed from 386 wastewater-farming households with two weeks recall period. The prevalence of diarrhea among children under the age of five years was 22.3%. Fig 2 depicts the prevalence of diarrhea among children under 5 years by age category. The odds of diarrhea were significantly associated with the age of the children (p<0.05). More than 75% of diarrhea occurrence was among children aged between 12 and 35 months, whilst children less than twelve months and between 37 to 59 months were less affected group. Children between 12 and 23 months and 24 and 34 months contribute 34.7% and 40.8% to the total odds of diarrhea under the age of five respectively (Fig 2).

### 3.3 Environmental characteristics

Among the major environment-related determinants that can potentially influence the prevalence of diarrhea, eight environmental factors with their strength of association were estimated using bivariate analysis (Table 2). The analysis showed that the type of toilet facility, availability

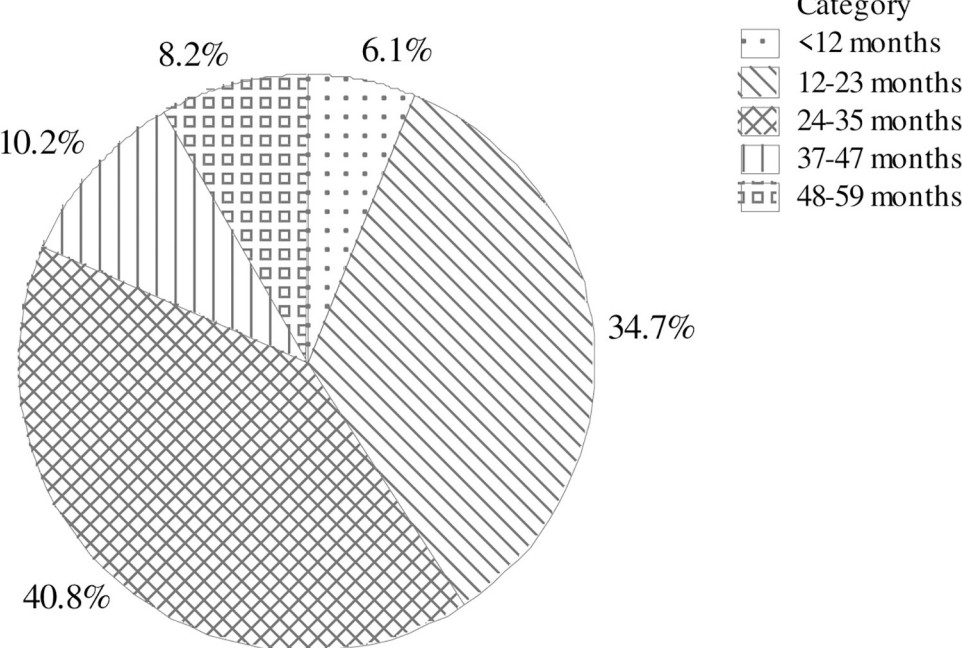

**Fig 2. Contributions of each age category to the overall occurrence of diarrhea.**

**Table 2. Univariate and Bivariate analysis of environmental /sanitation-related/ determinants of diarrhea among wastewater farming households.**

| Variables | Categories | Proportion | COR(95%CI) | P-value |
|---|---|---|---|---|
| Type of toilet facility | Flush | 25(6.5) | 1 | |
| | Pit latrine | 129(33.4) | 1.5 (0.3, 8) | 0.001 |
| | Composting | 232 (60.1) | 6(2.1, 11.7) | 0.001 |
| Water availability near the toilet | No | 332(86) | 4.8(1.1, 20.9) | 0.04 |
| | Yes | 54(14) | 1 | |
| Use of soap after toilet use | No | 175(85) | 1.5(0.6, 3.7) | 0.43 |
| | Yes | 15(32) | 1 | |
| Toilet distance from the house (m) | ≤5 | 342 (88.6) | 2.1(0.6, 7.5) | 0.2 |
| | >5 | 44 (11.4) | 1 | |
| Frequency of cleaning | Rarely | 20(5.2) | 6.4(0.8, 51.8) | 0.08 |
| | Daily | 80 (20.7) | 1 | |
| | Weekly | 187(48.4) | 2.9 (0.6, 14.2) | 0.2 |
| | Monthly | 99 (25.7) | 27(6.1, 119.6) | 0.0001 |
| Share toilet with other households | No | 161 (41.7) | 1 | |
| | Yes | 225 (58.3) | 4.5 (1.9, 10.5) | 0.001 |
| Defecation outside toilet | No | 26(54) | 1 | |
| | Yes | 74(154) | 9.6(1.3, 72 | 0.02 |

of hand washing water, toilet cleanliness, sharing toilet, and defecation outside the toilet has a significant association with the odds of child diarrhea. More than 90% of the households use pit latrines, whilst about 86% use composting toilets with no water for hand washing around the toilet. The use of composting toilet has 6 times more contribution to diarrhea compared to the flush toilet [(COR: 6.0, 95%CI (2.1, 11.7)], and the houses using pit latrine increases the odds of diarrhea by 1.5 compared to those households with flush latrine. The absence of hand-washing water around the toilet increases diarrhea by four compared to toilets with hand-washing water adjacent. Among 5% of the study population who rarely clean their toilet, diarrhea prevalence is five times higher compared to those who clean daily; and for those who clean once a month (25.7% of the study population), the odds of diarrhea prevalence is 27 times higher than those who clean daily.

## 3.4 Behavioral characteristics of farmers

Among a large number of hygiene-related determinants of diarrhea, six factors were found to be important and the bivariate analysis showed the presence of a significant association with the prevalence of diarrhea (p<0.05) (Table 3). About 97% of the respondents wash their hands before eating and feeding their children, which lowers diarrhea prevalence by five compared to the few respondents (3%) who do not wash their hands [COR: 5.8, 95% CI (1.6, 21.5)]. About 84% of the respondents regularly wash their hands after farm work, and regardless of the effect of other factors, the odds of child diarrhea was 21 times more likely among those who do not regularly wash their hands after farm work compared to those who do [COR: 21, 95%CI; (9.5, 47)]. Hand washing-related behaviors of the farmworkers such as the use of soap and onsite and offsite washing also was found to influence the odds of diarrhea (p<0.05). Furthermore, washing feet and boots before going back home from farm work also significantly influenced diarrhea cases (Table 3). Among those who wear boots during farm work, 85% do not wash their boots onsite, and thus the odds of diarrhea among their under-five children were 7.6 times higher than among those who wash their boots onsite.

**Table 3. Irrigation practice-related determinants of diarrhea prevalence among wastewater farming households.**

| Factors | Categories | Proportion (%) | COR(95%CI) | P-value |
|---|---|---|---|---|
| Hand washing before feeding children | No | 11(2.9) | 5.8(1.6, 21.5) | 0.008 |
| | Yes | 375 (97.2) | 1 | |
| Bathing after farm work | No | 275(75.2) | 5.3(1.8, 15.6) | 0.002 |
| | Yes | 111(28.8) | 1 | |
| Regular hand washing after work with soap | No | 61(15.8) | 21(9.5, 47) | 0.001 |
| | Yes | 325(84.2) | 1 | |
| On-site hand washing after work | No | 67(17.4) | 6.3(3,13) | 0.001 |
| | Yes | 319(82.6) | 1 | |
| Always wash feet after work | No | 70(18.2) | 31(11,88) | 0.001 |
| | Yes | 316(81.9) | 1 | |
| Boots/shoes wash after work | No | 329(85.2) | 7.6(1,57) | 0.05 |
| | Yes | 57(14.8) | 1 | |

## 3.5 Multivariable analysis of wastewater-related determinants of diarrhea

Socio-economic and demographic, environmental, and behavioral variables, which were found to be significant in the bivariate analysis at p-value ≤0.05 were further considered for multivariable regression (binomial multiple logistic regression) analysis. Table 4 shows the effect of wastewater-related determinants of diarrhea after adjusting for confounding factors.

**Table 4. Multivariable (binomial multiple logistic regression) analysis of wastewater—exposure determinants of under-five children diarrhea among wastewater irrigation households.**

| Name of variables | Category | Response | COR (96%CI), *P-value | AOR (95% CI), P-value |
|---|---|---|---|---|
| Wash with irrigation water | No | 106(27.5) | 6.2 (1.8, 20.7), 0.003 | 37.7 (3.1, 358), **0.004** |
| | Yes | 280 (72.5) | 1 | |
| Walking through the farm during irrigation | No | 134(37.7) | 1 | |
| | Yes | 252(52.3) | 3.6 (1.4, 8.9), 0.006 | 1.1 (0.08, 16.8), 0.9 |
| Cloth Contamination with the wastewater | No | 143(37.1) | 1 | |
| | Yes | 243(63.9) | 14 (3.3, 59.8), 0.000 | 10.8 (0.6, 205), **0.01** |
| Hand contamination with soil and irrigation water | No | 147(37) | 1 | |
| | Yes | 243(63) | 21(9.5, 47), 0.000 | 3.8(0.2, 89), 0.4 |
| Wearing protective wears during farming activities | Yes | 147(38.1) | 1 | |
| | No | 239 (62.9) | 12 (5.3, 27.5), 0.000 | 28.9 (3.9, 215), **0.001** |
| Wash vegetables with irrigation water | No | 224(58.1) | 1 | |
| | Yes | 162(41.9) | 6 (1.6, 23.2), 0.000 | 4.4 (0.8, 26.7), 0.1 |
| Hand washing with soap after work | No | 61(15.8) | 21 (9.5, 47), 0.000 | 26 (0.8, 812), **0.06** |
| | Yes | 325(84.2) | 1 | |
| Washing feet after farm work | No | 70(18.1) | 11.5 (5.6, 23.9), 0.000 | 1.4 (0.1, 17), 0.7 |
| | Yes | 316(81.9) | 1 | |
| Use working clothes at home | No | 87(22.6) | 1 | |
| | Yes | 298(77.4) | 54 (20.4, 143), 0.000 | 31.7 (4.4, 226), **0.001** |
| Onsite washing after work | No | 67(17.4) | 6.3 (3.1, 13.02), 0.000 | 1.1 (0.04, 30), 0.9 |
| | Yes | 319(82.6) | 1 | |
| Bring unwashed farm tools to home | No | 196(50.8) | 1 | |
| | Yes | 190(49.2) | 15.4(53, 44.9),0.000 | 94 (5.7, 157.5), **0.002** |

*P-value for all the variables during unadjusted odds ratio (COR) was less than 0.01 (P<0.01).

Among the multiple factors which re-introduce wastewater pathogens to households, washing bodies with irrigation water, cloth contamination, wearing protective clothes during irrigation activity, hand washing with soap after farm work, using working clothes at home, and bringing farm tools to home were found to significantly associated with the prevalence of child diarrhea (p<0.05).

About 72% of the respondents wash their body parts with irrigation water, and this caused the odds of diarrhea among childhood to be 37 times greater than diarrhea among those who don't use it for washing [(AOR: 37.7, 95%CI (3.1,358)]. About 77% of the farm workers use their working clothes at home, and this increased the odds of childhood diarrhea several times as compared to those who change their clothes at home [(AOR: 31.7, 95%CI (4.4, 226)]. Bringing farm tools home also increased the odds of child diarrhea by 94 times as compared to those who never took their tools home [AOR: 94, 95%CI (5.5, 1575)].

The prevalence of child diarrhea was 28.9 times higher in farming households that do not put on protective wear during farm works than those who use protective wears [AOR: 28.9, 95%CI(3.9,215)]. The other wastewater-irrigation-related factors such as walking through irrigation farm, hand contamination, washing vegetables with irrigation water, washing feet after farm work, and onsite hand washing less significantly to diarrhea in multivariate analysis, though the unadjusted odds ratio shows the presence of significant association (Table 4).

## 4. Discussion

The prevalence of diarrhea among children under the age of five years was 22.27%, which is in line with other reports. Several findings confirmed that the prevalence of diarrhea in wastewater irrigation areas is higher than in non-wastewater irrigation areas of the same environment [24].

In countries where wastewater irrigation is common practice, several researchers have reported the effect of wastewater irrigation on child diarrhea. Findings in Pakistan, Faisalabad 77% *giardia duodenalis* diarrhea and 5% other diarrheal infection among children between 2 and 12 ages and adult farmers [25], and in Morocco, Marrakesh 39% giardiasis (diarrhea), 28% amoebiasis and 21.34% salmonella infection among children between 2–14 years old [26].

In this study, the odds of diarrhea varied by sex of the farmworker and the children's age group. Children from female household heads were shown to be more likely to develop diarrhea than children from male household-heads. This may be associated with the fact that women typically have more proximity to their children than male counterparts, thereby increasing exposure of their children to wastewater-related contaminants. The majority of women take their children to the farm, where the children can play and have opportunity to touch everything around them.

Children under 12 months and over 37 months are less likely to develop diarrhea. The lower level of diarrhea among children under one year may be attributed to the fact that they are mostly under the mothers' close control for both movement and feeding; and children between 37 and 59 months of age have relatively good awareness what to touch, and what to eat, and some basic understanding of hygiene. Children of the later age learn what, when, and how to eat and to touch through direct experiences and by observing the behaviors of others [27]. However, children between ages 12 and 35 months are very active and eager to touch and take-to-mouth everything; primarily they learn by touching. Several assessment findings in different environmental settings show that children between 12 and 23 months of age are most vulnerable to diarrhea [28].

Among the sanitation-related variables, the type of toilet facility, availability of hand washing water, toilet cleanliness, sharing toilet and defecation outside the toilet significantly

contribute to the odds of diarrhea. A systematic review done globally about the impacts of hand-washing with soap on the odds of diarrheal disease reported that hand-washing with soap in community settings can reduce the risk of diarrheal disease by 42–47% [29]. The majority of the study households share toilets and thus the odds of diarrhea increased by 4.5 compared to those households having private toilets. Sharing toilet is a common lifestyle in areas where people live in congregated areas, and thus transmission of infectious/contagious diseases is higher. Evidence from 51 countries showed that shared sanitation is a risk factor for diarrhea although differences in socioeconomic status are important [30].

Regular hand and feet washing after work, washing boots, regular bathing significantly reduces diarrhea. They are effective strategies to interrupt the transmission of pathogens from a work to a domestic environment. Feeding children with dirty hands, not washing feet and shoes after work can be a means of introducing new pathogens or re-introducing the wastewater pathogens to the household environment. Several findings show that food contact with unwashed hands can be a source of diarrhea pathogens [31, 32]. Unwashed feet and shoes, particularly in poor sanitation environments, can contaminate the hands of a child and may lead hand to mouth transmission.

Occupational exposures such as using wastewater for washing clothes, feet and hands, wearing contaminated farm clothes inside home and bringing farm tools to home expose the whole family members to wastewater-related pathogens. This finding indicates that the odds of child diarrhea is higher among farm-worker' who do not use protective clothing compared to those wearing protective clothing. Several farmworkers wearing boots and hand gloves reduce their exposure to various wastewater contaminants. Several reports show that farm workers and their families are at high risk of wastewater-associated infection if they don't use appropriate type of protective clothing [33, 34]. The increased odds of child diarrhea among those who wear their working clothes at home, take their farm tools to home and touch their body parts with contaminated hands is because the farm workers can carry lots of contaminants and share everything on their body and clothes to their family members, particularly to their children. Reports also confirm that working clothes accumulate various types of microorganisms in large quantities and the potential 'take-home' exposure to these microorganism are of most concern for children and immunocompromised individuals [35].

Washing vegetables with irrigation water is a strong risk factor for the high prevalence of diarrhea among children in farming households. Vegetables are always contaminated with pathogenic microorganisms during growth, harvest, postharvest handling or distribution. Exposure to contaminated soils while harvesting and washing vegetables in wastewater are labelled as the major occupational risk attributing diarrhea FAO [11]. Previous reports show high levels of pathogenic organisms on vegetables produced from wastewater irrigation [36, 37]. Children may handle home-taken contaminated vegetables, which may contain diverse and large number of pathogens, like any other household items before washing. Moreover, consuming vegetables, particularly without proper washing during the recipe, is a common rout exposing consumers to various protozoal, viral and bacterial infections. Farmer's hand-washing practice can affect the whole family including children directly or indirectly. Several research findings show that caretakers' handwashing with soap significantly reduce the risk of diarrhea by 19%–53% [38, 39].

## 5. Conclusion

Childhood diarrhea among wastewater irrigation farming households in this study was considerably higher than has been reported previously. Diarrhea in children was strongly associated with a multitude of factors including hygiene, sanitation, and exposure to wastewater.

Exposures such as using wastewater for washing clothes, feet and hands, wearing contaminated farm clothes at home, wearing protective clothes, hand washing with soap after farm work, and bringing farm tools back to the home. While this study focused on diarrhea in young children, it is likely that entire households are exposed to wastewater-related pathogens, probably parasites. Unfortunately, without improved sanitation, these exposures can provide a source of reintroduction for wastewater pathogens back into the community. Thus, urban farmers need to think about alternative methods of irrigation or find ways to prevent contaminated wastewater reaching households where they can introduce infective gastrointestinal diseases. Further studies are therefore necessary to evaluate the health impact of chemical and microbial exposure to households living near wastewater irrigation sites.

## Supporting information

**S1 Data.**
(XLSX)

## Acknowledgments

We would like to thank Water and Land Resource Center of Addis Ababa University for their timely support. We wish to acknowledge the unreserved support of Agriculture officers in study area districts of Addis Ababa. We are also grateful to urban farmers for providing the requested from information.

## Author Contributions

**Conceptualization:** Adane Sirage Ali, Sirak Robele Gari, Bitew K. Dessie, Argaw Ambelu.

**Data curation:** Adane Sirage Ali.

**Formal analysis:** Adane Sirage Ali, Michaela L. Goodson.

**Funding acquisition:** Michaela L. Goodson, Claire L. Walsh.

**Investigation:** Adane Sirage Ali, Sirak Robele Gari.

**Methodology:** Adane Sirage Ali, Michaela L. Goodson, Bitew K. Dessie, Argaw Ambelu.

**Project administration:** Adane Sirage Ali, Bitew K. Dessie, Argaw Ambelu.

**Resources:** Adane Sirage Ali, Sirak Robele Gari, Claire L. Walsh.

**Software:** Adane Sirage Ali, Bitew K. Dessie.

**Supervision:** Sirak Robele Gari, Argaw Ambelu.

**Validation:** Sirak Robele Gari, Michaela L. Goodson, Claire L. Walsh, Bitew K. Dessie, Argaw Ambelu.

**Visualization:** Sirak Robele Gari, Claire L. Walsh.

**Writing – original draft:** Adane Sirage Ali.

**Writing – review & editing:** Adane Sirage Ali, Sirak Robele Gari, Michaela L. Goodson, Claire L. Walsh, Bitew K. Dessie, Argaw Ambelu.

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
