## [Decision Letter · Decision Letter 0]

10 Apr 2023

PONE-D-23-03536Prevalence and Risk Factors of Childhood Diarrhea among Wastewater Irrigating Urban Farming Households in Addis AbabaPLOS ONE

Dear Dr. Ali,

Thank you for submitting your manuscript to PLOS ONE. After careful consideration, we feel that it has merit but does not fully meet PLOS ONE’s publication criteria as it currently stands. Therefore, we invite you to submit a revised version of the manuscript that addresses the points raised during the review process.

We look forward to receiving your revised manuscript.

Kind regards,

Aiggan Tamene

Academic Editor

PLOS ONE

Additional Editor Comments:

Dear Author,

Firstly, I would like to thank you for submitting your manuscript to our publication. However, I regret to inform you that your manuscript requires major revisions before it can be considered for publication. While there are many strengths in the submission, there are several significant issues that need to be addressed before proceeding further. Elaborate on the revisions needed and include as much specific feedback as possible, such as areas that may require additional research, clarifications or restructuring of arguments, etc. If you choose to modify and resubmit your manuscript, I encourage you to take the necessary time to address our comments and suggestions thoroughly.

Some of the comments raised by the reviewers include, but are not limited to

1. [Issue 1, e.g. Clarification of the research question and better alignment of the objectives with the methodology]

2. [Issue 2, e.g. Expansion of the literature review to address earlier relevant studies and establish the context for your work]

3. [Issue 3, e.g. In-depth description of the data collection and analysis procedures to ensure reproducibility]

4. [Issue 4, e.g. Consideration of alternative explanations for the findings and a more nuanced interpretation of the results]

5. [Issue 5, e.g. Improvement of the manuscript's structure, organization, and clarity]

Reviewers' comments:

Reviewer's Responses to Questions

**Comments to the Author**

1. Is the manuscript technically sound, and do the data support the conclusions?

Reviewer #1: Partly

Reviewer #2: Partly

2. Has the statistical analysis been performed appropriately and rigorously? 

Reviewer #1: I Don't Know

Reviewer #2: Yes

3. Have the authors made all data underlying the findings in their manuscript fully available?

Reviewer #1: Yes

Reviewer #2: No

4. Is the manuscript presented in an intelligible fashion and written in standard English?

Reviewer #1: No

Reviewer #2: Yes

5. Review Comments to the Author

Reviewer #1: Materials and Methods: The sanitary situation of the study area (lines 105–111) may be deleted, and this can be discussed in the discussion. There is no sampling technique used to select the households. It is mentioned that the sample size was determined based on a ‘longitudinal study designed to investigate the incidence of diarrhea’, but the title and objective of the study specify that the prevalence of diarrhoea would be investigated; this is inconsistent and the study is not a longitudinal study. This study's design is also not mentioned explicitly. Furthermore, it is unclear how the researcher knew, prior to data collection, that the selected households had children under the age of 5. It appears that the questionnaire was pretested on the same population, which is not appropriate.

Results: Out of the 402 sample size, 386 responded; however, the total number of participants in some variables is different and inconsistent with the responded size in tables 1 and 2. It is preferable to write ‘p<0,05’ rather than ‘P<0.05’. In the results section, only findings should be given; there should be no discussion or citation of other studies. It is not clear how many under-5 children suffered from diarrhea out of the total number children studied in the previous two weeks.

Conclusion: The statement in lines 355–356 is not supported by the findings.

Reviewer #2: PONE D 2303536

In abstract

Please specify the method that you used for respondent selection

Introduction

Add the implication of your study to international readers and future research

Are the people in Addis Ababa drink from irrigation system? Is the wastewater-contaminate river water as the primary source of drinking water? How the irrigation water become the risk factor of diarrhea in that place? are people of Addis Ababa consumed raw vegetable from farming area that used wastewater irrigation? You should elaborate this information in introduction.

Add the previous study that claimed there are microbiological contamination in vegetable or drinking water sources in addis ababa. How many diarrhea cases in your study area ?

Inform the reader, your study respondent and why?

Material and method

Why you only choose two river (Big Akaki and little Akaki) as your study site?

I attach structured questionnaire in supplementary material section

Is there any Inclusion and exclusion criteria for your respondents?

Make sure you have number registration for STATA software.

Result and discussion

What are protective wears for farmworker that you mean in the result? why it is influence diarrhea cases in Addis Ababa? How children are exposed to diarrhea? Add some citation from the previous studies to discussion section!

Why walking through the farm during irrigation will influence diarrhea cases in children?

Based on your study result, what are your recommendation to prevent diarrhea? What your study implication for further research?

Please make sure your bibliography style in accordance with PLOSone journal style!

6. PLOS authors have the option to publish the peer review history of their article (what does this mean?). If published, this will include your full peer review and any attached files.

Reviewer #1: No

Reviewer #2: No

---

## [Author Response · Author response to Decision Letter 0]

25 May 2023

Response to comments contain three parts

- A. Revisions made as per journal requirements 

- B. Responses given to the academic editor’s comment

- C. Responses given to the reviewers’ comments 

A. Journal requirements

1. PLOS ONE - Style requirements

- File naming are modified following the PLOS ONE’s style

2. Professional language editors (two of the authors are native English speaker, from Newcastle University, UK. 

F Dr Claire Walsh – Co-author 

School of Engineering, Newcastle University (UK), Email: claire.walsh@ncl.ac.uk

Telephone: +44 (0)191 208 6647, Twitter: @clwnewc

https://www.ncl.ac.uk/engineering/staff/profile/clairewalsh.html

F Dr. Michaela Goodson – Co-author 

Email: michaela.goodson@newcastle.edu.my

Address: Newcastle University Malaysia

Jalan Sarjana 1, Educity@Iskandar

Iskandar Puteri, Johor, 79200

https://www.ncl.ac.uk/numed/people/staff-profiles/michaelagoodson.html

3. ORCID iD for the corresponding author 

I updated my profile information in the editorial manager. https://orcid.org/my-orcid?orcid=0000-0003-1962-1774

4. Your ethics statement should only appear in the Methods section of your manuscript. 

Ethical statement is included at the end of methods part, just after data analysis section. 

5. We note that Figure 1 in your submission contain [map/satellite] images which may be copyrighted. 

Figure 1, map of the study area is the map of the 10 sub-cities of Addis Ababa with overlaying rivers (SRTM 30x30 ArchGIS 10.8). We generated the map by GIS Software using map data and shapefiles within the public domain and with open access licenses from https://www.usgs.gov/centers/eros/science/usgs-eros-archive-digital-elevation-shuttle-radar-topography-mission-srtm-non

www.usgs.gov (https://www.usgs.gov/centers/eros/science/usgs-eros-archive-digital-elevation-shuttle-radar-topography-mission-srtm-non)

B. Response to the editor 

Dear Editor, first up, we express appreciation to your timely processing of our manuscript and your comments. We seriously considered yours and the reviewers’ comments. We hope that we addressed to all the clarifications, questions and comments forwarded in a given timeframe. We kindly request you to see the detail responses for your ‘5’ issues are answered together with the reviewers’ comments. 

1. [Issue 1, e.g. Clarification of the research question and better alignment of the objectives with the methodology]

Clarification is given in the introduction and methodology part. The research questions/problem statements are elaborated rewriting in an improved way and adding more points. 

2. [Issue 2, e.g. Expansion of the literature review to address earlier relevant studies and establish the context for your work]

We revised the introduction part as per the comments. Furthermore, additional references/earlier works in related areas are used for elaboration. 

3. [Issue 3, e.g. In-depth description of the data collection and analysis procedures to ensure reproducibility]

Based on the comments and suggestions given, the methodology part including the study design, sample size determination and sampling techniques are revised accordingly. 

4. [Issue 4, e.g. Consideration of alternative explanations for the findings and a more nuanced interpretation of the results]

Parts of the finding and interpretation which was not clear; and need better clarification, we made revisions. 

5. [Issue 5, e.g. Improvement of the manuscript's structure, organization, and clarity]

We made internal structural changes of the manuscript. We clarified unclear and less explained statements and ideas throughout the text. 

C. Response to Reviewers 

Response to reviewer #1

Thank you, dear reviewer. Giving fully consideration, we tried to respond to your comments one by one. 

Material and Method

The sanitary situation of the study area (lines 105–111) may be deleted, and this can be discussed in the discussion

 We shifted it to the discussion part accordingly.

There is no sampling technique used to select the households. 

Sampling technique is included in section 2.2. 

It is mentioned that the sample size was determined based on a ‘longitudinal study designed to investigate the incidence of diarrhea’, but the title and objective of the study specify that the prevalence of diarrhoea would be investigated; this is inconsistent and the study is not a longitudinal study. 

Correction is given. At the beginning, the sample size was determined for a longitudinal study, but later on, we changed it to a cross-sectional study. Now, we took the correction accordingly. 

This study's design is also not mentioned explicitly. 

- Now in this revision, we further elaborated the design at sections 2.2 and 2.3 

Furthermore, it is unclear how the researcher knew, prior to data collection, that the selected households had children under the age of 5. It appears that the questionnaire was pretested on the same population, which is not appropriate.

- We used an inclusion - exclusion criteria such as “only wastewater-irrigating farming households”, “only farming households having children under the age of 5 are included”. Thus, we included only those households who have children under the age of 5 are included in the study. The tools were pre-tested on those farming households who were not selected by simple random sampling. 

Results: Out of the 402 sample size, 386 responded; however, the total number of participants in some variables is different and inconsistent with the responded size in tables 1 and 2. 

- Yes, we corrected it. The major difference was typographical errors. Some missed values may also be in some variables, but statistically considered during analysis. 

 - We corrected all P-values to be written consistently.

It is preferable to write ‘p<0,05’ rather than ‘P<0.05’.

- Thank you very much for your comments. I think both ways are possible but I checked the PLOS ONE journal style, it follows the full-stop style, i.e. p<0.05 rather than p<0,05 

In the results section, only findings should be given; there should be no discussion or citation of other studies. 

- We revised it accordingly. 

It is not clear how many under-5 children suffered from diarrhea out of the total number children studied in the previous two weeks.

We revised it. We added a statement in the prevalence section, which an equivalent number for the 22.3% of the participants, i.e. approximately 86 children. 

Conclusion: The statement in lines 355–356 is not supported by the findings.

- We revised the statement. 

Response to reviewer #2 

Thank you, dear reviewer. Giving fully consideration, we hope that we addressed to all your clarifications and comments one by one. 

Abstract

Please specify the method that you used for respondent selection

We revised it by adding specific approaches we applied during study population selection. 

Introduction

Add the implication of your study to international readers and future research.

We added an elaboration as per the comments given.

Are the people in Addis Ababa drink from irrigation system?

- No, the river is severely polluted. However, those farming households along the rivers in addition to using the wastewater for irrigation, they use it for cloth washing, for bathing, washing vegetables and livestock drinking. 

Is the wastewater-contaminated river water as the primary source of drinking water? 

- No, it is not source of drinking water. 

How the irrigation water become the risk factor of diarrhea in that place? 

Wastewater pathogens can reintroduced into the households in several ways including washing bodies, clothes and vegetables with the irrigation water, cloth contamination, using working clothes at home, and bringing farm tools to home without onsite washing. Once the pathogens reach to the domestic environment, it can be ingested directly from hand to mouth, through food, plate and drinking water contamination and soon. Children at home touch the contaminated clothes, boots, farm materials and can contract the pathogens. 

Are people of Addis Ababa consumed raw vegetable from farming area that used wastewater irrigation? You should elaborate this information in introduction.

Yes. The wastewater irrigation supplies more than 90% of the leafy vegetables and more than 65% of all types of vegetables. Those leafy vegetables such as lettuce and chard, which normally eaten raw, are produced from the wastewater irrigation and sold everywhere in the city. 

Add the previous study that claimed there are microbiological contamination in vegetable or drinking water sources in addis ababa. 

We added literatures related with this part. 

How many diarrhea cases in your study area? 

We included in section 3.2. 

Inform the reader, your study respondent and why?

We revised the problem statement part in the introduction section. 

Material and method

why you only choose two rivers (Big Akaki and little Akaki) as your study site?

They are the only rivers in the city, cross the city from north to south. They receive all types of pollutants from multiple sources including households, hotels, restaurants, government institutes, industries, hospitals and other health service sectors. Although the rivers are severely polluted, they are intensively used for vegetable production, livestock drinking, cloth washing, construction etc. Therefore, they are becoming a threat to the public health. 

Attach structured questionnaire in supplementary material section.

The questionnaire can be given as supplementary material 

Is there any Inclusion and exclusion criteria for your respondents?

Yes, we have. Now, in this revised version we added inclusion criteria at section 2.2 

Make sure you have number registration for STATA software.

Yes, I was using licensed software. 

Result and discussion

What are protective wears for farmworker that you mean in the result?

 Protective wears include those wears protecting the farm workers’ body and clothes from wastewater during occupation. These include boots, plastic protective coating, masks etc.

Why it is influence diarrhea cases in Addis Ababa? 

The third highest diarrhea prevalence in the country is found in Addis Ababa. WASH condition in Addis Ababa is very poor. Sanitation condition is very poor (indicated in the discussion part). There is no full water supply coverage in the city, particularly in the slum areas. Almost all the city water supply is intermittent. These conditions are usually associated with increased occurrence of diarrhea. 

How children are exposed to diarrhea? Add some citation from the previous studies to discussion section!

We revised accordingly. Included in the discussion part. 

Why walking through the farm during irrigation will influence diarrhea cases in children.

During irrigation activity, most farmers without any protective wears walk across the farm to channelize the irrigation water, which causes direct contamination of the foot and clothes. Moreover, walking through splashes the irrigation water and contaminate the farmers’ cloth. There is no onsite body washing practice in all of the farm workers, which implies that the farmers go home with their contaminated body and clothing and reintroduce wastewater pathogens to domestic environment, where children normally touch and manipulate everything. 

Based on your study result, what are your recommendation to prevent diarrhea? 

 Our recommendation spans around awareness creation/ education and improving WASH conditions among the farming households. 

What your study implication for further research?

As a way forward, ways to reduce wastewater-related pathogen loads such as by employing onsite wastewater treatment need to be considered.

Please make sure your bibliography style in accordance with PLOSone journal style!

We changed the bibliography from ‘numbered’ style to ‘Vancouver superscript bracket’ style.

---

## [Decision Letter · Decision Letter 1]

22 Jun 2023

PONE-D-23-03536R1Prevalence and Risk Factors of Childhood Diarrhea among Wastewater Irrigating Urban Farming Households in Addis AbabaPLOS ONE

Dear Dr. Ali,

Thank you for submitting your manuscript to PLOS ONE. After careful consideration, we feel that it has merit but does not fully meet PLOS ONE’s publication criteria as it currently stands. Therefore, we invite you to submit a revised version of the manuscript that addresses the points raised during the review process.

We look forward to receiving your revised manuscript.

Kind regards,

Aiggan Tamene

Academic Editor

PLOS ONE

Journal Requirements:

Reviewers' comments:

Reviewer's Responses to Questions

**Comments to the Author**

1. If the authors have adequately addressed your comments raised in a previous round of review and you feel that this manuscript is now acceptable for publication, you may indicate that here to bypass the “Comments to the Author” section, enter your conflict of interest statement in the “Confidential to Editor” section, and submit your "Accept" recommendation.

Reviewer #1: (No Response)

Reviewer #2: All comments have been addressed

2. Is the manuscript technically sound, and do the data support the conclusions?

Reviewer #1: Yes

Reviewer #2: Yes

3. Has the statistical analysis been performed appropriately and rigorously? 

Reviewer #1: Yes

Reviewer #2: Yes

4. Have the authors made all data underlying the findings in their manuscript fully available?

Reviewer #1: (No Response)

Reviewer #2: Yes

5. Is the manuscript presented in an intelligible fashion and written in standard English?

Reviewer #1: (No Response)

Reviewer #2: Yes

6. Review Comments to the Author

Reviewer #1: Minor changes are required to the updated document.

The author(s) revised ‘p<0.05’ but some ‘P<0.05’ yet to be revised.

In the text, the reference number should be cited sequentially, such as [6].

Materials and Methods:

It is stated that twelve urban framing sites were chosen for data collection. It should be mentioned that how many of the twelve urban framing sites, were wastewater-irrigated farming sites and how many were non-wastewater-irrigated farming sites?

The calculated sample size was 407, although it is mentioned that data was collected from 402 households, with responses obtained from 386 of them.

Results:

In the title of Table 1, ‘occurrence of diarrhea’ should be included.

In the tables, the number of responses against each variable is not comparable; therefore, it is preferable to mention the total number of responses against each variable.

Conclusion:

It was indicated that the households may be exposed to toxic chemicals; this should be discussed in the discussion before making any comments in the conclusion.

Reviewer #2: (No Response)

7. PLOS authors have the option to publish the peer review history of their article (what does this mean?). If published, this will include your full peer review and any attached files.

Reviewer #1: No

Reviewer #2: **Yes: **RATNA DWI PUJI ASTUTI

---

## [Author Response · Author response to Decision Letter 1]

25 Jun 2023

All the comments given are properly addressed. 

 - We made necessary revisions as per the comments given by changing “P<0.05” to “p<0.05”. 

 - References numbers are updated and corrected throughout the paper.

 - We corrected the typographic error in sample size

 - We revised the table as per the comments given by adding " occurrence of diarrhea". 

 - the errors on table are also corrected. 

 - We made the necessary amendments in the conclusion part.

---

## [Editor Report · Decision Letter 2]

29 Jun 2023

Prevalence and Risk Factors of Childhood Diarrhea among Wastewater Irrigating Urban Farming Households in Addis Ababa

PONE-D-23-03536R2

Dear Dr. Ali,

We’re pleased to inform you that your manuscript has been judged scientifically suitable for publication and will be formally accepted for publication once it meets all outstanding technical requirements.

Kind regards,

Aiggan Tamene

Academic Editor

PLOS ONE
---

## [Editor Report · Acceptance letter]

4 Jul 2023

PONE-D-23-03536R2 

Prevalence and Risk Factors of Childhood Diarrhea among Wastewater Irrigating Urban Farming Households in Addis Ababa 

Dear Dr. Ali:

I'm pleased to inform you that your manuscript has been deemed suitable for publication in PLOS ONE. Congratulations! Your manuscript is now with our production department. 

Kind regards, 

on behalf of

Mr Aiggan Tamene 

Academic Editor

PLOS ONE